# Precipitation before Flowering Determined Effectiveness of Leaf Removal Timing and Irrigation on Wine Composition of Merlot Grapevine

**DOI:** 10.3390/plants10091865

**Published:** 2021-09-09

**Authors:** Runze Yu, Matthew W. Fidelibus, James A. Kennedy, Sahap Kaan Kurtural

**Affiliations:** 1Department of Viticulture and Enology, University of California, Davis, 1 Shields Ave., Davis, CA 95616, USA; crzyu@ucdavis.edu (R.Y.); mwfidelibus@ucanr.edu (M.W.F.); 2Department of Viticulture and Enology, California State University, Fresno, 2360 E. Barstow Ave. M/S VR 89, Fresno, CA 93740, USA; 3Functional Phenolics, LLC, 5305 River Rd, North Ste B, Keizer, OR 97303, USA; j.a.kennedy@outlook.com

**Keywords:** canopy management, deficit irrigation, vineyard mechanization, flavonoids, hot climate viticulture

## Abstract

Grapevine productivity, and berry and wine flavonoid concentration, depend on the interactions of cultivar, environment, and applied cultural practices. We characterized the effects that mechanical leaf removal and irrigation treatments had on the flavonoid concentration of ‘Merlot’ (*Vitis vinifera*, L.) grape berries and wines in a hot climate over two growing seasons with contrasting precipitation patterns. Leaves were removed by machine, either at prebloom (PBLR), or at post-fruit-set (PFLR), or not removed (control) and irrigation was either applied as sustained deficit irrigation (SDI) at 0.8 of crop evapotranspiration (ET_c_) from budbreak to fruit set, or regulated deficit irrigation (RDI) at 0.8 ET_c_ from bud break to fruit set, 0.5 ET_c_ from fruit set to veraison, and 0.8 ET_c_ from veraison to harvest, of ET_c_ In 2014, PFLR reduced the leaf area index (LAI) compared to control. The RDI decreased season-long leaf water potential (Ψ_Int_) compared to SDI. However, in 2015, none of the treatments affected LAI or Ψ_Int_. In 2014, berry flavonoid concentrations were reduced by PBLR as well as SDI. SDI increased the flavonoid concentrations in wine, and PFLR increased some wine flavonols in one season. No factor affected the concentrations of wine proanthocyanidins or mean degree of polymerization. Thus, mechanical PFLR and RDI may increase berry flavonoid accumulation without yield reduction, in red wine grapes cultivars grown in hot climates when precipitation after bud break is lacking. However, spring precipitation may influence the effectiveness of these practices as evidenced by this work in a changing climate.

## 1. Introduction

The San Joaquin Valley (SJV) of California is a major wine grape growing region of the United States. In 2019, it produced 48% of the total wine grapes crushed in the state of California [1]. The average grower return for Merlot wine grapes from the SJV was only USD 310 per ton, whereas the state average for that variety was USD 826 per ton. Red wine grapes from this region are generally priced lower than similar grapes from cooler growing regions because its climate, specifically the high growing season temperatures and rapid growing degree day accumulation, favor high yields of fruit with relatively low berry flavonoid concertation at harvest. Hence, wines made of SJV grapes are usually marketed as high volume, low-cost wines. The economics of this industry favor the development of production practices that minimize grape and wine production cost while maintaining or improving grape and wine quality. For example, mechanization of canopy management practices and the implementation of optimal irrigation practices can minimize labor costs and improve grape berry flavonoid concentration [2,3,4,5].

Flavonoids are critical in determining the color, flavor, and mouthfeel of red wine [6] and thus directly affect wine quality [7]. They are also the primary antioxidants that help plants cope with environmental stresses. Their biosynthesis and concentration respond to environmental cues, including water deficits, solar radiation exposure, and temperature [8,9,10]. Previous studies observed that moderately increasing the severity of water deficits and solar radiation increased the content of two major flavonoid classes, anthocyanins and flavonols, in berries [8,11]. However, excessive exposure of grape berries to sunlight, high air temperatures, and water deficits, reduces these compounds at harvest [10,12]. Another major class of flavonoids, proanthocyanidins, determine wine astringency [13] and help stabilize wine color via copigmentation [14]. They are less sensitive to environmental stresses compared to other flavonoids [15]. However, under relatively severe environmental stress, even proanthocyanidin composition and concentrations may be altered, in a manner similar to the other flavonoids [16,17,18]. 

Leaf removal and deficit irrigation are the two cultural practices most commonly used to manage canopy structure and plant water status, other than dormant pruning. Leaf removal in the fruiting zone can directly affect canopy microclimate, and thereby affect berry flavonoid accumulation [3]. Leafing may affect grapevine source-sink relations, which would also contribute to the changes in berry development [19,20]. Removal of leaves around clusters of grapes at different developmental stages were investigated to help growers understand the various benefits selective leaf removal can provide. When leaves were removed early (prebloom), berry set, and therefore yield, were reduced in cool climate vineyards [21,22,23]. Studies in warm and hot climates deduced that early leaf removal increased berry total soluble solids (TSS), and berry skin flavonoid concentration without adversely affecting yield [3,24]. When leaves were removed later in the season in cool climates, the total proanthcoyanidin content was increased in berries, but decreased in wine [21]. Other studies suggested that late leaf removal could enhance berry anthocyanins [3,25]. 

Water is a critical environmental factor for grapevine physiological development [26]. Water deficits reduce grapevine vegetative growth [27] and berry weight [28]. Severe water deficits might inhibit photosynthesis [29] and promote berry maturity and vine dormancy by stimulating abscisic acid biosynthesis [30], Mild to moderate water deficits improve berry chemical composition due, in part, to suppressing grapevine vegetative growth, and thereby increasing the sink strength of berries [31]. Moreover, water deficits increase berry flavonoid concentrations [32], and the increases in TSS and flavonoid concentrations can be attributed to the alteration of biosynthetic pathways [8,33], or simply the reduction in berry weight due to water loss [34,35]. 

Imposing water deficits on grapevines at different developmental stages can result in different effects. The SJV in California is a semiarid region and growers typically replace 70% to 80% of crop evapotranspiration from bud-break to harvest [2,35]. Preveraison water deficits increased berry anthocyanin concentration whereas post veraison water deficits promoted TSS concentration [32]. Castellarin et al. (2007) reported that preveraison water deficits hastened sugar accumulation and anthocyanin biosynthesis [8], where the genes related to anthocyanin biosynthesis, including flavonoid 3-hydroxylase (F3H), di-hydroflavonol 4-reductase (DFR), UDP-glucose: flavonoid 3-*O*-glucosyl-transferase (UFGT) and glutathione S-transferase (GST) were upregulated. There was also evidence showing that both pre and postveraison water deficits can enhance anthocyanin biosynthesis [36]. Sometimes water deficits may increase the total anthocyanin content while the extractable anthocyanins might be lower [23]. Postveraison water deficits may also limit flavonoid biosynthetic accumulation [29]. Nevertheless, berry dehydration due to water deficits can overrule the metabolomic regulation and directly determine the flavonoid concentration in wine.

We previously studied the effects of mechanical leaf removal and water deficits on the anthocyanin content and profile of grape berries [3]. In this study, we subjected grapevines in a hot climate to mechanical canopy management treatments and water deficits in an attempt to promote flavonoid accumulation in grapes and wine. We hypothesized that the leaf removal and water deficit would improve the berry and wine flavonoid profile without adversely affecting yield. Overall, the objective of this study was to investigate the physiological and chemical impacts of mechanical leaf removal and water deficits on berry and wine flavonoid concentration of Merlot in a hot climate. 

## 2. Results

### 2.1. Weather at the Experiment Site

The vineyard received approximately 34% more precipitation in the 2014–2015 postharvest season (from September to April) than it did during the same time period in 2013–2014 (Figure 1A). The vineyard received little precipitation during the growing seasons, with only 1.4 mm and 10.2 mm of precipitation in 2014 and 2015, respectively. In the second season, precipitation prior to anthesis was higher than in 2014, and the GDD was also greater, 1711.8 °C GDD in 2015, versus 1590 °C, in 2014 (Figure 1B). However, due to greater early season precipitation, in 2015, Ψ_leaf_ did not reach the water deficit threshold of −1.0 MPa to initiate irrigation until fruit set (Table 1). Due to the higher temperature from June to harvest, the ET_c_ was higher in the second season, where 1151.1 L per vine were applied comparing to 1098 L per vine in 2014. Additionally, the applied water amounts of SDI were slightly different between two seasons, where there were 1483.0 L per vine applied in 2014 and 1423.6 L per vine applied in 2015.

### 2.2. Canopy LAI and Season-Long Plant Water Status

Canopy LAI and season-long Ψ_Int_ were assessed in both seasons (Figure 2). In 2014, PFLR had significantly lower LAI when compared to Control (Figure 2A1). The LAI of grapevines subjected to PBLR was not different from Control, or PFLR, whereas in 2015, vines subjected to either LR treatment had similar LAI (Figure 2A2). Overall, vines had higher LAI in the second season compared to the first season. Season-long Ψ_Int_ were −1.3 MPa with RDI in 2014, which was significantly lower than the −1.1 MPa of SDI (Figure 2B1). However, in 2015, there was no significant difference in Ψ_Int_ between the treatments (Figure 2B2). We did not detect a significant interaction between leaf removal and irrigation on LAI or Ψ_Int_ in either year of the experiment. 

### 2.3. Yield Components and Berry Composition

The yield results of the 2014 trial were previously reported [3]. In 2015, vines subjected to PBLR had the fewest clusters and lowest yield and average berry weight (Table 2). Vines subjected to RDI had fewer clusters than other vines. Irrigation treatments did not affect other yield components. Neither treatment affected the leaf area to fruit ratio in 2015. The berries were harvested at the similar maturity stages in 2014 and 2015. The TSS in 2015 was affected by the leaf removal treatments; where PRLR showed the highest TSS, PFLR showed the lowest. 

### 2.4. Berry Skin Flavonoid Concentration—Anthocyanins and Flavonols

The concentration of berry skin flavonoids was measured in 2014 and 2015 (Table 3). In 2014, berries from vines subjected to PBLR had less delphinidin, cyanidin, and petunidin compared to berries from vines subject to the other two leaf removal treatments. The di-hydroxylated anthocyanins were the highest in berries from vines subjected to PFLR. In 2015, however, PFLR obtained the highest concentrations of malvidin and tri-hydroxylated anthocyanins. It also obtained the highest concentrations of quercetin, myricetin, and total flavonols in the second season. In 2014, RDI increased delphinidin, cyanidin, petunidin, tri-hydroxylated, and total anthocyanin. However, there was no difference in either anthocyanins or flavonols between SDI and RDI in 2015. When comparing the two years, the flavonoid concentrations in the second year were generally lower than the first.

### 2.5. Wine Flavonoid Concentration

Wine flavonoids were measured in 2014 and 2015, and differences observed in berry skins did not transfer into wine with leaf removal treatments (Table 4). In 2014 and 2015, there was no differences observed with leaf removal treatments in any of the anthocyanin derivatives. However, the differences in flavonols from leaf removal treatments were significant enough to be observed in wine, where PFLR had higher quercetin, myricetin, and total flavonols although there was no separation between PBLR and PFLR in 2015. As for irrigation treatments, in 2014, SDI increased the concentrations of cyanidin and petunidin, and also increased quercetin and total flavonols in both seasons. Like berry skin flavonoid concentrations, the concentrations of most of the wine anthocyanin and flavonol derivatives were lower in the second season than the first one. 

The wine proanthocyanidin concentration and composition were measured in 2014 and 2015 (Table 5). There were no significant differences in any of the proanthocyanidin subunits due to either leaf removal or deficit irrigation treatments, except epicatechin (EC) terminal subunits, which were reduced by PBLR in 2014. The general concentrations, including the total proanthocyanidins, were lower in the second season compared to the first season, and the mDP was higher. 

## 3. Discussion

### 3.1. Grapevine Canopy and Water Status

Leaf removal can reduce canopy leaf area [37,38], as occurred with PFLR in 2014. Previous studies indicated grapevine canopies would regrow when leaves were removed early in the season [19,22], which might partly explain why PBLR did not affect LAI. The second season had more precipitation between bud break to anthesis, which promoted vegetative growth, as observed from the generally higher LAI values in 2015. These results were corroborated by the relatively higher average Ψ_Int_ of −1.0 MPa in the second season comparing it to the first season, which was −1.2 MPa. Our results provided further evidence that leaf removal can be more effective in altering canopy size in more arid seasons, presumably because the drier soil made it easier to control canopy size by induced water deficits [39]

Application of the RDI treatment decreased Ψ_Int_ in 2014, which indicated that the 30% reduction of applied water amounts between fruit set and verasion was sufficient to alter the season-long plant water status [2]. However, the second season did not show such a separation in season-long plant water status. We attributed this to the precipitation received before budbreak in 2015, which delayed the attainment of moderate water deficit stress as the Ψ_leaf_ did not reach −1.0 MPa until fruit set.

### 3.2. Yield Components and Berry Composition

The effects of various timings of leaf removal on berry development and composition were previously investigated [21,38,40,41]. Late (post-fruit-set) leaf removal can affect grapevine yield and berry composition [38,42], but prebloom leaf removal has been shown to be more effective in modifying yield and berry composition than post-fruit-set leaf removal [21,43]. In our study, yield was reduced by PFLR in 2014, and by PBLR in 2015. This inconsistency might be due to treatment effects on LAI. Vines subjected to PFLR had the lowest LAI in 2014, perhaps sufficiently to reduce yield capacity compared to vines with larger canopies, as typically found in SJV [44]. Previous studies reported that berry weight may be affected by leaf removal, especially when the leaf removal was conducted early in the season [19,37]. Skin weight was affected by leaf removal, where altered canopy microclimate by leaf removal could be the direct factor to manipulate berry skin weight [22,45]. In our study, berry skin weight was reduced with PFLR in 2014. This might be because that late leaf removal diminished the growth of berry skin, as witnessed in previous studies [22,46]. 

Previous studies showed that leaf removal increased berry TSS concentration [19,47]. Leaf removal could increase berry TSS by dehydration sunlight [48,49], or increased carbohydrate accumulation and partitioning to the fruits [19]. However, in our study, treatments had few and small effects on TSS. 

### 3.3. Berry and Wine Flavonoids

Previous studies investigated the effects of leaf removal on grape berry skin anthocyanin and flavonol concentration, and some studies focused on scrutinizing the various outcomes from the different timings of leaf removal [40,42]. However, in our case, PFLR was not effective in increasing berry skin anthocyanin concentration in either season. Previous work indicated that berry exposure to solar radiation late in the season might make the berries more prone to negative effects of radiation exposure and higher air temperature [24]. As for flavonols, they are generally reported to be sensitive to solar radiation [50,51], but we did not notice a difference in flavonols in 2014 even though the LAI was significantly reduced by PFLR. The berry weight might have been the determining factor, where PFLR did not significantly reduce berry weight, hence it did not increase the concentration either. When comparing the first season to the second, the anthocyanins and flavonols were generally lower, although the TSS at harvest in both years were at the same level. Previous studies reported that <0.8–1.2 m^2^ of leaf area per kg of fruits could inhibit berry maturation [52]. The second season had a lower leaf area to fruit ratio—the plants did not have sufficient canopies as source tissues to reach the same maturity in both TSS and flavonoids, which might have contributed to the lower flavonoid accumulation in 2015.

RDI significantly reduced plant water status and increased berry anthocyanin concentrations in 2014. Previous studies had shown that deficit irrigation could increase berry anthocyanin concentration [8,33]. In our case, most of the anthocyanin derivatives were greater with RDI. As for flavonols, our study did not indicate that reduction of applied water amounts with RDI had noticeable influences on berry flavonols in either year. This agreed with previous work, that flavonols are relatively insensitive towards water deficits [8]. 

In our study, only a portion of the significant treatment effects on anthocyanins and flavonols were carried into wine. We attributed these discrepancies to differences in berry skin extractability affected by both treatments. Previous work indicated that higher permeability of the skin cell walls would lead to more advanced maturity, eventually increasing the extractability of flavonoids [53]. PBLR showed the ability to promote berry maturity (i.e., TSS) in 2014, and the more advanced maturity might have diminished flavonoid concentration benefiting from the leaf removal treatment. Some research attributed this observance to the warm and hot climate, where the impacts of leaf removal and deficit irrigation might be unhelpful in such regions, to a point that berry flavonoids are not increased, or are even decreased [10,54]. 

Among the three classes of flavonoids, proanthocyanidins are most the chemically stable and less easily manipulated by cultural practices or grapevine physiological status [55,56,57]. In our study, there were minimal effects from the treatments, where only EC terminal subunits were significantly affected. Some previous studies were able to see significant effects, mainly positive, of sun exposure and water deficits on berry or wine proanthocyanidin concentration [15,16,58]. However, in warm/hot climates, environmental stresses could be sufficiently severe to degrade berry proanthocyanidins [59,60]. This might have contributed to the lower total proanthocyanidin concentrations in the second season compared to the first due to the higher air temperatures in 2015. Among the proanthocyanidin subunits, the EGC extension subunits were the most drastically reduced in 2015. This was corroborated by previous work, where EGC extension subunits were sensitive towards air temperature [61]. 

## 4. Materials and Methods

### 4.1. Site Description

The experiment was conducted at a commercial vineyard in Stanislaus County, CA, USA. Merlot grapevines (clone FPS 01) grafted to Freedom (27% *V. vinifera* hybrid) rootstock were planted in 1998 in 2.13 m × 3.35 m (vine × row) spacing, in rows oriented North–South. The grapevines were head-trained and supported with a California sprawl trellis which consisted of a cordon wire at 1.37 m above vineyard floor, and two foliage wires separated by a 20 cm t-top. The grapevines were cane-pruned to six canes with eight nodes each. The vineyard was drip-irrigated with pressure-compensating emitters spaced at 1.1 m with two emitters per vine delivering 2 L/h each. 

### 4.2. Experimental Design

The experiment was a three (leaf removal) × two (deficit irrigation) arranged factorially with a split-plot design with four replicated blocks. Three rows comprised one block and four buffer rows separated each block. The main plot was the leaf removal treatments, the subplot was irrigation treatments. Each experimental unit consisted of 285 vines, and 48 vines were selected, which were sampled and measured during the growing season.

### 4.3. Mechanical Leaf Removal Treatments

In 2014 and 2015, two leaf removal treatments were applied: a prebloom leaf removal treatment (PBLR), a post-fruit-set leaf removal treatment (PFLR), and an untreated control (Control). The leaf removal treatments were applied mechanically on the east side of the canopy with a roll-over type leaf remover (Model EL-50, Clemens Vineyard Equipment Inc., Woodland, CA, USA). A 50 cm window in the fruiting zone was created after the treatment. PBLR was applied at 200 GDD in 2014 and 2015. PFLR was applied at 644 GDD and 600 GDD in 2014, and 2015, respectively. 

### 4.4. Irrigation Treatments and Weather

The amount of water to apply each week, crop evapotranspiration (ET_c_), was calculated as the product of reference evapotranspiration (ET_o_) and crop coefficient (K_c_) [62]. The reference ET_o_, air temperature, and precipitation were obtained from the California Irrigation Management Information System (CIMIS) weather station (#206) in Denair, CA. For crop coefficient calculation, a neighboring row was irrigated to 100% of ET_o_ replacement. The shade cast under 24 plants in this row then was measured to calculate percent shaded area to calculate the crop coefficient weekly. The crop evapotranspiration was then estimated as described by Williams and Ayars (2005) [63]. A sustained deficit irrigation (SDI) at 0.8 ET_c_ was applied weekly from anthesis until harvest. A regulated deficit irrigation (RDI) treatment was applied at 0.8 ET_c_ from anthesis to fruit set, 0.5 ET_c_ from fruit set to veraison, and back to 0.8 ET_c_ from veraison until harvest. The growing degree days (GDD) were calculated with the air temperature acquired from the CIMIS station as GDD = [(T_max_ + T_min_)/2 − T_ref_], where T_max_ was the maximum air temperature, T_min_ was the minimum air temperature, and T_ref_ was the base temperature 10 °C. All other cultural practices were carried out according to University of California guidelines for the area. GDD calculation for both years only considered the time prior to harvest. 

### 4.5. Plant Water Status Assessment

Leaf water potential (Ψ_leaf_) of the grapevines was monitored weekly. Four sun-exposed leaves were measured with the use of a pressure chamber (Model 610 Pressure Chamber Instrument., PMS Instrument Co., Corvallis, OR, USA) as previously reported elsewhere by Cook et al. (2015) [3]. To summarize the season-long plant water status, Ψ_leaf_ integrals were calculated by using natural cubic splines. The values were then divided by the number of the days between the first and the last Ψ_leaf_ water measurements in each year to make the data comparable to each individual measurement as Ψ_Int_.

### 4.6. Leaf Area Index and Yield Components

Leaf area was determined at 50% veraison from 24 vines per experimental unit. Four random shoots were collected from the east and west sides of the canopy per vine. The leaves were removed from the shoots, and the leaf area was measured with a leaf area meter (LI-3100C, LI-COR Biosciences, Lincoln, NE, USA). Total leaf area for each vine was calculated as the average leaf area per shoot multiplied by the average shoot numbers per vine. Lastly, leaf area index (LAI) was calculated as the ratio between the leaf area to the ground surface area for each vine (7.14 m^2^). 

Yield components were measured on a single harvest date as the berry TSS reached 24 °Brix in each year. All clusters in each treatment replicate were picked, counted, and weighed to determine the number of clusters per vine, average cluster weight, and yield per vine. Two sets of samples were collected, including one set of 100 berries to assess average berry mass, and another set of 20 berries to assess the dry skin mass and further skin flavonoid analysis. 

### 4.7. Chemicals

All chromatographic solvents were of HPLC grade. Acetonitrile, acetone, ascorbic acid, ethanol, glacial acetic acid, maleic acid, methanol, potassium metabisulfite, potassium hydroxide, and sodium hydroxide were purchased from Fisher Scientific (Santa Clara, CA, USA). Phloroglucinol, (−)-epicatechin (EC), and hydrochloric acid were purchased from Sigma-Aldrich (St. Louis, MO, USA). Malvidin-3-*O*-glucoside and quercetin-3-*O*-rutinoside were purchased from Extrasynthése (Genay, France). Dihydrogen ammonium phosphate and phosphoric acid were purchased from VWR (Visalia, CA, USA). Hydrochloric acid and sodium acetate anhydrous were purchased from E. M. Science (Gibbstown, NJ, USA) and Mallinckrodt (Phillipsburg, NJ, USA), respectively.

### 4.8. Berry Composition

The first set of 100 berries were crushed and the juice was used for the analysis of berry primary metabolites, including TSS, titratable acidity (TA), and must pH. Must TSS was measured (as °Brix) with a digital refractometer (Atago PR-32, Atago CO., Ltd., Bellevue, WA, USA). The TA was measured by titrating the must to an endpoint pH of 8.2 with 0.1N sodium hydroxide on an endpoint titrator (Mettler-Toledo DL15, Mettler-Toledo International Inc., Columbus, OH, USA). Must pH was measured by a glass electrode pH meter (Accumet™ AB15, Fisher Scientific, Pittsburg, PA, USA).

### 4.9. Extraction of Skin Flavonoids

Berry skins were manually removed from the second berry set of 20 berries, and lyophilized with a centrivap (Centrivap Benchtop Centrifugal Vacuum Concentrator 7810014 equipped with Centrivap −105 °C Cold Trap 7385020, Labconco, Kansas City, MO, USA). Dry skin masses were recorded after lyophilization, and then extracted in 20 mL 66% (*v*·*v*^−1^) acetone solution in the dark for 24 h. Acetone extracts were vacuum filtered, solids were discarded, and 1 mL of liquid was collected. The acetone in the extracts was removed with the Centrivap, and the solution left was brought up to 5 mL with water. Samples were then centrifuged for 15 min at 1400× *g*, and the supernatant was filtered by PTFE membrane filters (diameter: 13 mm, pore size: 0.45 μm, VWR, Seattle, WA, USA), and transferred into High Performance Liquid Chromatography (HPLC) vials before analysis.

### 4.10. Berry and Wine Flavonoid Analysis

Skin and wine anthocyanins and flavonols were analyzed by a reversed-phase HPLC system (Agilent 1100 series, Santa Clara, CA, USA) equipped with a system controller, a vacuum degasser (Model: G1379A), a quaternary pump (Model: G1311A), an autosampler, a thermostatted column compartment (Model: G1316A), and a DAD/UV-vis detector (Model: G1315A). A C18 column (LiChrosphere 100 RP-18, 4 × 520 mm^2^, 5 mm particle size, Agilent Technologies, Santa Clara, CA, United States) was used as previously reported elsewhere by Yu et al., (2016). 

The concentrations for wine proanthocyanidin subunits were assessed by acid catalysis in the presence of excess phloroglucinol (phloroglucinolysis) by reversed-phase HPLC using the same instrument as mentioned above [64]. To purify proanthocyanidins, DSC-18 solid phase extraction (SPE) cartridges (bed weight: 500 mg, volume: 6 mL, Sigma-Aldrich, St. Louis, MO, USA) were used. Briefly, the SPE column was preconditioned with 3 column volumes of methanol and then with 3 column volumes of water. We passed 1 mL of samples through the column and washed it by 3 column volumes of water to remove impurities. Then, the resided sample was eluted with 3 × 3 mL of methanol. The eluent was lyophilized to powder and then re-dissolved in 1 mL of methanol and ready for phloroglucinolysis. 

For phloroglucinolysis, 0.25 mL eluent was mixed with 0.25 mL of phloroglucinolysis reagent (100 g·L^−1^ phloroglucinol and 20 g·L^−1^ ascorbic acid with 0.2 N hydrochloric acid in methanol). The proanthocyanidin of interest in the mixture solution was reacted at 50 °C in water bath for 20 min. Then, the reaction was stopped by mixing 200 μL of the mixture solution with 1 mL of stopping reagent (40 mM aqueous sodium acetate) and then directly transferred into HPLC vials. A column with two Chromolith RP-18e (100 × 4.6 mm^2^) columns serially connected was used, and it was protected by a guard column with the same material (4 × 4 mm^2^) from EM Science (Gibbstown, NJ, USA). The mobile phase flow rate was 3.0 mL·min^−1^, and two mobile phases were used, which included solvent A = 1% aqueous acetic acid (*v*·*v*^−1^) and solvent B = 1% acetic acid in acetonitrile (*v*·*v*^−1^). The HPLC flow gradient started with 97% A with 3% B, 82% A, 18% B at 14 min, 20% A, 80% B at 14.01 min, 97% A, 3% B at 16.01 min until 20 min. The compound identification and quantification were conducted by using ChemStation version B.04.03 with the use of peak area measurements at 280 nm for all proanthocyanidin subunits. The standard used was (-)-epicatechin (Sigma-Aldrich, St. Louis, MO, USA).

### 4.11. Winemaking

We carried out microscale fermentations in 2014 and 2015. The fruits from the 24 vines were hand-harvested, and the 4 replicates of total 14 kg of fruits per treatment-replicate were used for fermentation. The fruits were crushed and destemmed by using a crusher–destemmer (Cantinetta C.d.A., ZAMBELLI Enotech, Camisano Vicentino, Italy), potassium metabisulfite was added to the must (50 mg·kg^−1^ SO_2_). The must from each treatment was then fermented in a 4 L vessel according to Sampaio et al. [65]. Briefly, each vessel was equipped with a Teflon cap, a fermentation airlock, and a food-grade polyethylene screen to keep must caps submerged in juice. Each lot was inoculated with 0.2 g·L^−1^ of commercial yeast, Saccharomyces cerevisiae Meyen ex Hansen (Cotes des Blancs, Red Star Yeast Prod. Oakland, CA, USA). All fermentations were carried out indoors with temperatures maintained at 23 °C. Punch-downs were carried out twice a day, where the polyethylene screens were submerged, until the alcoholic fermentation was completed. The fermentation progress was monitored by a hydrometer until dryness. The wines were then pressed with a vacuum pump (MaximaDry™, Fisher Scientific, Waltham, MA, USA) with a pressure of 0.2 MPa maintained for 30 min, the crudes were removed by filter (P8, diameter: 11.0 cm, Fisher Scientific, Waltham, MA, USA) placed in a Buchner funnel (CoorsTek 60242, Golden, CO, USA). Potassium metabisulfite was added to the wine to retain the SO_2_ level at 50 mg·kg^−1^ and cold stabilized at −2 °C. Then, the wines were bottled in 375 mL glass bottles with screw caps. 

### 4.12. Statistical Analysis

Interactions between year and treatments were tested and, whenever these interactions were significant (*p* ≤ 0.05), analysis was conducted separately for each year. The results were subjected to a two-way (leaf removal × irrigation) analysis of the variance (ANOVA) using in R (version 1.1.442, RStudio, Inc., Boston, MA, USA) appropriate for split–split plot with a factorial arrangement of treatments. All data were tested for normality using Shapiro–Wilk’s test, some data required a combination of log and square root transformations where deemed necessary in 2014 and 2015. Treatment means were considered significantly different by Tukey’s honestly significant difference adjustment at *p* ≤ 0.05.

## 5. Conclusions

Leaf removal and water deficits have been extensively studied in viticultural research. However, there is a need to better understand the effects of different timings of mechanical leaf removal and deficit irrigation on grape berry and wine flavonoid concentration in hot climates where majority of the world’s wine grapes are grown. Wine grape growers in hot climate regions must make up for relatively low prices, due to the low flavonoid concentration in grape berries/wine, with high yields of fruit without further compromising fruit quality. Thus, we studied two factors, leaf removal and deficit irrigation, to determine their effects on berry and wine flavonoid concentrations, to see if they may be useful to wine growers in hot climates. To conclude, we have determined PFLR and RDI in hot climates may increase flavonoid concentration in red wine grape berries but possibly not large enough of an effect to beneficially affect wine flavonoid concentration. Additionally, our study provides evidence on the feasibility of mechanical leaf removal and water deficits on the berry and wine quality improvement in large-acreage commercial vineyards in a hot climate where precipitation prior to anthesis is a determining factor of the effectiveness of cultural practices. Future work in the region may consider relating the soil water content, precipitation to anticipated fruit and wine composition. 

## Figures and Tables

**Figure 1 plants-10-01865-f001:**
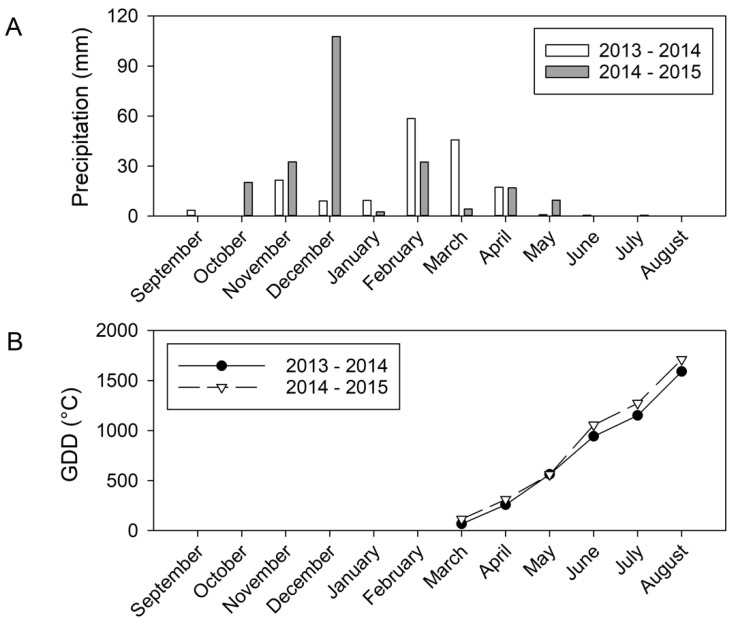
Weather at the experiment site in 2014 and 2015, acquired from California Irrigation Management Information System (CIMIS) station (#206, Denair, CA). (**A**) Monthly precipitation. Note: GDDs were calculated until August, when the harvests in both years occurred. (**B**) Growing degree day accumulation starting in March of each year.

**Figure 2 plants-10-01865-f002:**
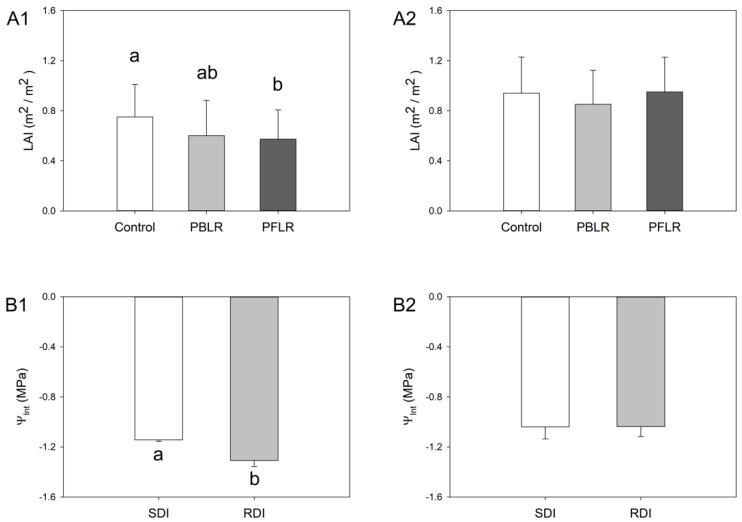
Grapevine leaf area index (LAI) and season-long leaf water potential integrals (Ψ_Int_) as affected by mechanical leaf removal treatments and deficit irrigation treatments, respectively. (1) 2014, (2) 2015, (**A**) LAI by mechanical leaf removal treatments, including untreated control (Control), prebloom leaf removal (PBLR), and post-fruit-set leaf removal (PFLR), (**B**) Ψ_Int_ by deficit irrigation, including sustained deficit irrigation (SDI) and regulated deficit irrigation (RDI). Columns with different letters are significantly different at *p* < 0.05 according to Tukey’s HSD.

**Table 1 plants-10-01865-t001:** Crop evapotranspiration (ET_c_) and applied water amount between two irrigation regimes in a Merlot vineyard in Denair, California in 2014 and 2015.

	2014				2015			
	SDI		RDI		SDI		RDI	
	ET_c_ ^a^ (mm)	Applied Water (L/vine)	ET_c_ ^a^ (mm)	Applied Water (L/vine)	ET_c_ ^a^ (mm)	Applied Water (L/vine)	ET_c_ ^a^ (mm)	Applied Water (L/vine)
bud break-fruit set	29.72	181.16	29.72	181.16	n/a ^b^	n/a ^b^	n/a ^b^	n/a ^b^
fruit set-veraison	122.19	862.20	72.37	477.70	101.64	726.57	71.18	454.11
veraison-harvest	61.49	439.60	61.49	439.60	97.50	696.99	97.50	696.99
total	213.40	1482.96	163.58	1098.46	199.13	1423.55	168.68	1151.09

^a^ SDI utilized 80% ET_c_ from bud-break to harvest, for RDI utilized 80% ET_c_ from bud-break to fruit set, 50% ET_c_ from fruit set to veraison, and 80% ET_c_ from veraison to harvest. ^b^ Irrigation was not applied before the leaf water potential reached −1 MPa. Hence, there was no water applied prior to fruit set in 2015.

**Table 2 plants-10-01865-t002:** Yield components and berry primary metabolites at harvest in a Merlot vineyard in Denair, California in 2015 ^a,b,c^.

		Components of Yield						Berry Composition		
		Cluster/Vine	Yield (kg)/Vine	Average Berry wt. (g)	Skin Mass (mg)	Skin to Berry Weight Ratio (%)	Leaf Area to Fruit Ratio (m^2^/kg)	TSS (°Brix)	pH	TA (g/L)
2014										
LR	Control	55 ^a^	6.2 ^a^	1.09	45.3 ^a^	4.15 ^a^	0.94	24.3	3.60	4.83
	PBLR	54 ^a^	6.1 ^a^	1.07	42.9 ^ab^	4.00 ^a^	0.87	24.1	3.62	4.66
	PFLR	45 ^b^	4.5 ^b^	1.11	39.5 ^b^	3.56 ^b^	0.94	24.2	3.64	4.69
	*p* value	**	*	ns	*	*	ns	ns	ns	ns
IRRI	SDI	52	6.1 ^a^	1.14 ^a^	42.7	3.83	0.79 ^b^	23.9 ^b^	3.63	4.83
	RDI	51	5.3 ^b^	1.04 ^b^	42.3	3.83	1.05 ^a^	24.5 ^a^	3.61	4.62
	*p* value	ns	ns	**	ns	ns	*	ns	ns	ns
LR × IRRI		ns	*	ns	ns	ns	ns	ns	ns	ns
2015										
LR	Control	120 ^a^	16.6 ^a^	1.30 ^a^	36.5	2.69	0.43	24.6 ^ab^	3.48	7.53
	PBLR	105 ^b^	13.4 ^b^	1.27 ^b^	37.4	2.92	0.49	24.9 ^a^	3.50	7.90
	PFLR	118 ^ab^	15.1 ^ab^	1.32 ^ab^	32.3	2.46	0.47	24.1 ^b^	3.49	7.44
	*p* value	*	**	*	ns	ns	ns	**	ns	ns
IRRI	SDI	120 ^a^	15.7	1.31	36.8	2.81	0.46	24.3	3.48	7.73
	RDI	108 ^b^	14.4	1.33	34.0	2.57	0.47	24.8	3.50	7.52
	*p* value	**	ns	ns	ns	ns	ns	ns	ns	ns
LR × IRRI		ns	ns	ns	ns	ns	ns	ns	ns	ns

^a^ ANOVA to compare data (*p* value indicated); Letters within columns indicate significant mean separation according to Tukey’s honestly significant difference test at *p* value ≤ 0.05, where “*”: *p* value ≤ 0.05; “**”: *p* value ≤ 0.001. ^b^ LR: leaf removal; IRRI: irrigation; PBLR: prebloom leaf removal; PFLR: post-fruit-set leaf removal; ns: not significant. ^c^ A portion of this table was previously published © 2015 American Society for Enology and Viticulture AJEV 66:266–278.

**Table 3 plants-10-01865-t003:** Grape berry skin flavonoid concentration at harvest in a Merlot (*Vitis vinifera* L.) vineyard in Denair, California in 2014 and 2015 ^a,b,c^.

		Anthocyanins								Flavonols		
		2014										
		Delphinidin	Cyanidin	Petunidin	Peonidin	Malvidin	Tri-hydroxylated	Di-hydroxylated	Total anthocyanins	Quercetin	Myricetin	Total flavonols
LR	Control	11.06 ^a^	6.56 ^a^	11.94 ^a^	17.32	94.05	117.05	23.88 ^ab^	140.93	13.33	0.73	14.06
	PBLR	8.32 ^b^	5.45 ^b^	9.57 ^b^	15.56	85.70	103.59	21.01 ^b^	124.60	13.09	0.85	13.94
	PFLR	11.19 ^a^	7.25 ^a^	12.01 ^a^	18.86	90.52	113.72	26.11 ^a^	139.83	15.00	0.88	15.88
	*p* value	**	*	**	ns	Ns	ns	*	ns	ns	ns	ns
IRRI	SDI	9.28 ^b^	6.73	10.43 ^b^	18.08	81.71	101.43 ^b^	24.81	126.24 ^b^	13.54	0.80	14.34
	RDI	11.10 ^a^	6.11	11.91 ^a^	16.39	98.56	121.57 ^a^	22.50	144.07 ^a^	14.09	0.84	14.93
	*p* value	*	ns	*	ns	**	**	ns	**	ns	ns	ns
LR × IRRI		ns	ns	ns	ns	Ns	ns	ns	ns	ns	ns	ns
		2015										
LR	Control	7.89	5.29	8.92	16.67	70.46 ^b^	87.27 ^b^	21.96	109.23	4.89 ^b^	1.71 ^b^	7.81 ^b^
	PBLR	8.47	5.65	9.35	18.28	70.88 ^ab^	88.70 ^b^	23.93	112.64	6.35 ^ab^	1.88 ^b^	9.69 ^ab^
	PFLR	8.91	5.76	10.33	18.44	85.96 ^a^	105.20 ^a^	24.21	129.40	7.41 ^a^	2.36 ^a^	11.53 ^a^
	*p* value	ns	ns	ns	ns	*	*	ns	ns	**	*	**
IRRI	SDI	8.19	5.60	9.31	17.63	74.46	91.96	23.24	115.19	6.17	1.96	9.58
	RDI	8.66	5.53	9.76	17.96	77.08	95.49	23.49	118.98	6.27	2.02	9.78
	*p* value	ns	ns	ns	ns	Ns	ns	ns	ns	ns	ns	ns
LR × IRRI		ns	ns	ns	ns	Ns	ns	ns	ns	ns	ns	ns
Year		**	*	*	ns	***	***	ns	**	***	***	***
Year × LR		*	ns	ns	ns	Ns	ns	ns	ns	ns	ns	ns
Year × IRRI		ns	ns	ns	ns	*	ns	ns	ns	ns	ns	ns
Year × LR × IRRI		ns	ns	ns	ns	Ns	ns	ns	ns	ns	ns	ns

^a^ ANOVA to compare data (*p* value indicated); Letters within columns indicate significant mean separation according to Tukey’s honestly significant difference test at *p* value ≤ 0.05, where “*”: *p* value ≤ 0.05; “**”: *p* value ≤ 0.001; “***”, *p* value ≤ 0.0001. ^b^ All compounds were expressed as mg per kg of berry fresh weight. ^c^ Abbreviations: LR: leaf removal; IRRI: irrigation; PBLR: prebloom leaf removal; PFLR: post-fruit-set leaf removal; SDI: sustained deficit irrigation; RDI: regulated deficit irrigation, ns: not significant.

**Table 4 plants-10-01865-t004:** Wine flavonoid concentration in a Merlot (*Vitis vinifera* L.) vineyard in Denair, California in 2014 and 2015 ^a,b,c^.

		Anthocyanins								Flavonols		
		2014										
		Delphinidin	Cyanidin	Petunidin	Peonidin	Malvidin	Tri-hydroxylated	Di-hydroxylated	Total anthocyanins	Quercetin	Myricetin	Total flavonols
LR	Control	3.98	1.86	8.19	2.11	128.23	140.40	3.97	144.37	19.56	6.70	26.26
	PBLR	4.23	1.92	8.58	2.12	135.87	148.68	4.04	152.72	23.66	7.60	31.26
	PFLR	4.44	2.25	9.31	2.42	124.73	138.49	4.68	143.16	24.05	7.67	31.72
	*p* value	ns	ns	Ns	ns	ns	ns	ns	ns	ns	ns	ns
IRRI	SDI	3.81	1.85 ^b^	7.80 ^b^	2.15	122.82	134.43	4.00	138.43	19.44 ^b^	6.59	26.03 ^b^
	RDI	4.62	2.17 ^a^	9.59 ^a^	2.29	136.40	150.61	4.46	155.07	25.40 ^a^	8.06	33.46 ^a^
	*p* value	ns	*	*	ns	ns	ns	ns	ns	*	ns	*
LR × IRRI		ns	ns	Ns	ns	ns	ns	ns	ns	ns	ns	ns
		2015										
LR	Control	7.15	2.04	6.14	8.60	71.00	84.29	10.64	94.93	9.68 ^b^	3.71 ^b^	13.39 ^b^
	PBLR	8.37	2.06	6.59	9.09	70.59	85.56	11.15	96.71	11.82 ^ab^	4.24 ^ab^	16.05 ^ab^
	PFLR	9.31	2.55	7.67	10.35	85.21	102.20	12.89	115.09	12.71 ^a^	4.72 ^a^	17.43 ^a^
	*p* value	ns	ns	Ns	ns	ns	ns	ns	ns	*	**	*
IRRI	SDI	7.64	2.21	6.60	9.03	72.58	86.82	11.24	98.06	10.16 ^b^	3.97 ^b^	14.13 ^b^
	RDI	8.91	2.22	7.01	9.66	78.63	94.54	11.89	106.43	12.64 ^a^	4.48 ^a^	17.12 ^a^
	*p* value	ns	ns	Ns	ns	ns	ns	ns	ns	*	*	*
LR × IRRI		ns	ns	Ns	ns	ns	ns	ns	ns	ns	ns	ns
Year		***	ns	**	***	***	***	***	***	***	***	***
Year × LR		ns	ns	Ns	ns	ns	ns	ns	ns	ns	ns	ns
Year × IRRI		ns	ns	Ns	ns	ns	ns	ns	ns	ns	ns	ns
Year × LR × IRRI		ns	ns	Ns	ns	ns	ns	ns	ns	ns	ns	ns

^a^ ANOVA to compare data (*p* value indicated); Letters within columns indicate significant mean separation according to Tukey’s honestly significant difference test at *p* value ≤ 0.05, where “*”: *p* value ≤ 0.05; “**”: *p* value ≤ 0.001; “***”, *p* value ≤ 0.0001. ^b^ All compounds were expressed as mg per L. ^c^ Abbreviations: LR: leaf removal; IRRI: irrigation; PBLR: prebloom leaf removal; PFLR: post-fruit-set leaf removal; SDI: sustained deficit irrigation; RDI: regulated deficit irrigation; ns: not significant.

**Table 5 plants-10-01865-t005:** Wine proanthocyanidin subunit concentration in a Merlot (*Vitis vinifera* L.) vineyard in Denair, California in 2014 and 2015 ^a,b,c^.

		Extension Subunits				Terminal Subunits		Total Proanthocyanidins	mDP
		EGC	C	EC	ECG	C	EC		
		2014							
LR	Control	125.34	28.30	303.18	8.52	88.13	136.29 ^a^	689.76	3.09
	PBLR	121.63	26.79	275.97	8.56	83.36	113.30 ^b^	629.62	3.21
	PFLR	123.49	28.67	286.42	7.86	92.06	128.50 ^ab^	667.43	3.04
	*p* value	ns	Ns	ns	ns	ns	*	ns	ns
IRRI	SDI	123.93	29.44	291.60	7.58	88.84	132.45	673.84	3.06
	RDI	124.06	26.37	286.31	9.09	87.18	120.19	652.63	3.15
	*p* value	ns	Ns	ns	ns	ns	ns	ns	ns
LR × IRRI		ns	Ns	ns	ns	ns	ns	ns	ns
		2015							
LR	Control	44.30	19.51	214.10	29.81	70.43	47.47	425.62	3.65
	PBLR	49.41	20.29	229.99	31.80	71.52	49.02	452.03	3.79
	PFLR	58.89	20.66	237.69	31.30	72.54	48.05	469.12	3.91
	*p* value	ns	Ns	ns	ns	ns	ns	ns	ns
IRRI	SDI	46.35	19.07	217.74	29.55	69.24	47.38	429.34	3.70
	RDI	56.39	21.39	238.84	32.62	74.05	49.12	472.41	3.88
	*p* value	ns	Ns	ns	ns	ns	ns	ns	ns
LR × IRRI		ns	Ns	ns	ns	ns	ns	ns	ns
Year		***	***	***	***	***	***	***	***
Year × LR		ns	Ns	ns	ns	ns	ns	ns	ns
Year × IRRI		ns	Ns	ns	ns	ns	ns	ns	ns
Year × LR × IRRI		ns	Ns	ns	ns	ns	ns	ns	ns

^a^ ANOVA to compare data (*p* value indicated); Letters within columns indicate significant mean separation according to Tukey’s honestly significant difference test at *p* value ≤ 0.05, where “*”: *p* value ≤ 0.05; “***”, *p* value ≤ 0.0001. ^b^ All compounds were expressed as mg per L. ^c^ Abbreviations: LR: leaf removal; IRRI: irrigation; PBLR: prebloom leaf removal; PFLR: post-fruit-set leaf removal; SDI: sustained deficit irrigation; RDI: regulated deficit irrigation; ns: not significant; C: (+)-catechin; EC: (-)-epicatechin; ECG: (-)-epicatechin-3-*O*-gallate; EGC: (-)-epigallocatechin; mDP: mean degree of polymerization. mDP was calculated as the ratio of total proanthocyanidins to the terminal subunits.

## Data Availability

The raw data supporting the conclusions of this article will be made available by the authors, without undue reservation.

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
