# Peer review of "Precipitation before Flowering Determined Effectiveness of Leaf Removal Timing and Irrigation on Wine Composition of Merlot Grapevine"

_plants, 2021, doi:10.3390/plants10091865_

Round 1
Reviewer 1 Report
Dear Sirs,My comments are in the 2 files attached. One is the manuscript with my comments in balloons and the other one is a word file with the most remarkable things that should be correct, go deep.
And the contents of word file are as follows:
Summary
This work deals on the effects of two cultural practices, leaf removal and irrigation on berry and wine flavonoids in a vineyard in Central Valley (California USA). The trial develops for 2 seasons and results are determined by the rainfall pattern of each season. The authors base this study on that leaf removal and water deficti improved flavonoids in other viticultural areas, now they want to know when is the best moment to practice leaf removal and to impose the water déficit. Their results show little differences between treatments and they are not consistent from one year to another. The most significant effect is the “year” effect through the seasonal rainfall distribution and yield.
Format aspects
Material and Methods should be placed before the Results section.
References
Many references are referred with a number and some other with the name of the first author and year. Use the journal format for the whole document.
Tables
Table 1. I wonder if what you name as ETc in SDI is really 0.8·ETc and what you name ETc in the RDI is 0.5 or 0.8·ETc depending on the phenological stage. For example, If I divided total water applied in 2014 in SDI, 1482.96 by plant spacing (7.15 m2) to convers to mm, this turns out 207 mm which is similar to the left column value, 213mm.
Table 2. For he same variable, decimals sometimes are 2 and sometimes 1. When there is not statistic differences, sometimes you write “ns” and sometimes the P-value. Choose the same format for the all parameters in all tables.
Tables 3, 4 and 5. You present data as mean ± SD which makes tables difficoult to read. I know this is the correct way to present results but as the way to see them is less attractive and in fact this variability is already taken into account in the ANOVA to decide if it is significant/no-significant, I suggest to remove the ± SD from tables.
p-value: I personally think that is better seen and to express ANOVA results as *, ** and ns for significant at P=0.05, 0.001 and no-significant respectively. To see the real p-value with so many decimals is confusing and tables are more difficoult to see and to get conclusions.
Peonidin 2014 results are wrongly writen.
Material and Methods
It is very well written. I have just added some considerations in balloons relative to how often some measurements were made, what is the time span between PBLR and PFLR, to clarify better how is Ψint calculated, and so on.
I have to highlight that I have not reviewed the methodology of the flavoid extraction as I am not an expert on this field.
Results
Comments in balloons in the pdf file. They focus on clarifying things rather than correct things.
Discussion
Grapevine canopy and water status. I would remark that the second year there is not difference in LAI nor plant water status. This means that there should not be found differences because of the treatments but other causes such us yield, berry size, and so on. This is what results show in 2015.
Discussion in general focus on the differences between PBLR and PFLR and irrigation strategies but there are not references to CONTROL. Many times there are not differences between PFLR and control, so that I guess if LR at any time make sense under this situation where growers try to reduce costs. In this case the cost is time, machine and labour.
Berry and Wine flavonoids. You focussed discussion on differences between treatments but I would recommend authors to turn it the other way round, that is, to remark those aspects where differences do not appear. I see there are more parameters with no treatment effect than those with positive/negative effect on either on berry or wine. Both years are diferent no becasue of the treaments but because of the rainfall distribution along the season. So, at this point, I think this research is to recommend growers something regarding when practising the leaf removal and how much water to apply. One answer could be “we should go on longer time to check the results for 2-3 more years”. Another answer is to recommend something base on a study of the rainfall distribution for the last 10 years and look into how frequent are this quantity of rainfalls from budbreak to fruitset and conclude that the X% of the years leaf removal will not have effect on berry not wine composition so growers may benefith of saving money by no practicing any leaf removal.
On this way, I see your results on wine are no-different between treaments any year, so … why to recommend leaf removal if eventually it will not turn out in any beneficial on wine composition?
I have write some more comments in the pdf file for you to clarify things.

Author Response
Material and Methods should be placed before the Results section.
A: We thank reviewer #1 for this comment on the format, we followed the format guide from MDPI plants and the “materials and methods” section is recommended to be placed at the last.
References
Many references are referred with a number and some other with the name of the first author and year. Use the journal format for the whole document.
A: The references were fixed to be uniformly presented in numerical order.
Tables
Table 1. I wonder if what you name as ETc in SDI is really 0.8·ETc and what you name ETc in the RDI is 0.5 or 0.8·ETc depending on the phenological stage. For example, If I divided total water applied in 2014 in SDI, 1482.96 by plant spacing (7.15 m2) to convers to mm, this turns out 207 mm which is similar to the left column value, 213mm.
A: ETc in this table did not represent 100%. ETc for SDI was 80% and for RDI was 80%-50%-80%. But we added the denotation to avoid such confusion.
Table 2. For the same variable, decimals sometimes are 2 and sometimes 1. When there is not statistic differences, sometimes you write “ns” and sometimes the P-value. Choose the same format for the all parameters in all tables.
A: We thank reviewer #1’s extensive attention, we unified the digits for these variables.
Tables 3, 4 and 5. You present data as mean ± SD which makes tables difficoult to read. I know this is the correct way to present results but as the way to see them is less attractive and in fact this variability is already taken into account in the ANOVA to decide if it is significant/no-significant, I suggest to remove the ± SD from tables.
p-value: I personally think that is better seen and to express ANOVA results as *, ** and ns for significant at P=0.05, 0.001 and no-significant respectively. To see the real p-value with so many decimals is confusing and tables are more difficoult to see and to get conclusions.
A: We modified the way we present the p values.
Peonidin 2014 results are wrongly writen.
A: We correctly it according to reviewer #1’s advice.
Material and Methods
It is very well written. I have just added some considerations in balloons relative to how often some measurements were made, what is the time span between PBLR and PFLR, to clarify better how is Ψint calculated, and so on.
I have to highlight that I have not reviewed the methodology of the flavoid extraction as I am not an expert on this field.
Results
Comments in balloons in the pdf file. They focus on clarifying things rather than correct things.
A: We thank reviwer #1’s comments, we addressed all the questions in this cover letter and made the modifications in the re-submitted file.
Discussion
Grapevine canopy and water status. I would remark that the second year there is not difference in LAI nor plant water status. This means that there should not be found differences because of the treatments but other causes such us yield, berry size, and so on. This is what results show in 2015.
A: We thank reviewers #1’s comment, we actually discussed this in the flavonoid discussion section already.
Discussion in general focus on the differences between PBLR and PFLR and irrigation strategies but there are not references to CONTROL. Many times there are not differences between PFLR and control, so that I guess if LR at any time make sense under this situation where growers try to reduce costs. In this case the cost is time, machine and labour.
A: In 2014 this is true, but from our second-year data PFLR and SDI increased some flavonoid derivatives significantly. Thus, there are still potential benefits by utilizing PFLR. However, the effects did not show up in the wine, but we addressed that by stating the treatments did very little effect on wine flavonoid concentration.
Berry and Wine flavonoids. You focussed discussion on differences between treatments but I would recommend authors to turn it the other way round, that is, to remark those aspects where differences do not appear. I see there are more parameters with no treatment effect than those with positive/negative effect on either on berry or wine. Both years are diferent no becasue of the treaments but because of the rainfall distribution along the season. So, at this point, I think this research is to recommend growers something regarding when practising the leaf removal and how much water to apply. One answer could be “we should go on longer time to check the results for 2-3 more years”. Another answer is to recommend something base on a study of the rainfall distribution for the last 10 years and look into how frequent are this quantity of rainfalls from budbreak to fruitset and conclude that the X% of the years leaf removal will not have effect on berry not wine composition so growers may benefith of saving money by no practicing any leaf removal.
A: We thank reviewer #1’s recommendations, however, that will go beyond the purpose of this study. We addressed that information for growers in our other publications.
On this way, I see your results on wine are no-different between treaments any year, so … why to recommend leaf removal if eventually it will not turn out in any beneficial on wine composition?
A: We thank reviewers #1’s comment, we have rephrased on conclusion to avoid such confusion. We stated leaf removal can improve berry flavonoid concentration, but not so much in wine. But still, this study can provide evidence on the effectiveness of mechanical leaf removal on wine flavonoid concentration in a hot climate.
Reviewer 2 Report
The investigation of environmental effects on grape and wine composition is a really interesting topic, that will direct researchers and stakeholders (even farmers) toward a more sustainable viticulture. In the present study, grapes sampled during two seasons with contrasting precipitation patterns allowed the characterization of the effects of timings of mechanical leaf removal and irrigation. The experimental design is correct to answer the unsolved question and even the results are supported by valuable statistics. On my opinion, the paper can be accepted with minor revisions. Please check the legend of figures and tables, some of them seems uncomplete. There are too many tables in the paper and not all the data are discussed, please change some of them in graphs and add the complete dataset as supplementary material. Please add the ID of Merlot clones used in this study.Author Response
Reviewer #2:
The investigation of environmental effects on grape and wine composition is a really interesting topic, that will direct researchers and stakeholders (even farmers) toward a more sustainable viticulture. In the present study, grapes sampled during two seasons with contrasting precipitation patterns allowed the characterization of the effects of timings of mechanical leaf removal and irrigation. The experimental design is correct to answer the unsolved question and even the results are supported by valuable statistics. On my opinion, the paper can be accepted with minor revisions. Please check the legend of figures and tables, some of them seems uncomplete. There are too many tables in the paper and not all the data are discussed, please change some of them in graphs and add the complete dataset as supplementary material. Please add the ID of Merlot clones used in this study.
A: We thank reviewer #2’s comments, we added the ID of Merlot clones used in this study. However, we did not convert any tables into graphs, we modified some of the tables to make them easier to read. The trial was designed as a factorial and it is difficult to read the graphs with that many lines. We hope this is acceptable
Reviewer 3 Report
The paper describes a research on leaf removal and water deficit as combined factors on Merlot grapevine. The topic is quite original, as it matches both leaf removal and water relations; there are several papers concerning these two topics separately but a factorial form of the two is new as far as I know. This is very important as it adds new knowledge to the grapevine management. The paper is well written, the text is clear and easy to read. The research is well set, material and methods are clear and results and discussion are appropriate. I only suggest to put the references in the text following the direction of the editorial board, because in some cases are cited by number and in some other are cited extensively. Just a correction in Line 38 (concentration).
Author Response
The paper describes a research on leaf removal and water deficit as combined factors on Merlot grapevine. The topic is quite original, as it matches both leaf removal and water relations; there are several papers concerning these two topics separately but a factorial form of the two is new as far as I know. This is very important as it adds new knowledge to the grapevine management. The paper is well written, the text is clear and easy to read. The research is well set, material and methods are clear and results and discussion are appropriate. I only suggest to put the references in the text following the direction of the editorial board, because in some cases are cited by number and in some other are cited extensively. Just a correction in Line 38 (concentration).
A: We thank reviewer #3’s comments, we have corrected the reference format in the context. And we made the correction in line 38.